# Association between Parkinson’s Disease and Cancer: New Findings and Possible Mediators

**DOI:** 10.3390/ijms25073899

**Published:** 2024-03-31

**Authors:** Andrei Surguchov, Alexei A. Surguchev

**Affiliations:** 1Department of Neurology, Kansas University Medical Center, Kansas City, 3901 Rainbow Boulevard, Kansas City, KS 66160, USA; 2Department of Surgery, Section of Otolaryngology, Yale School of Medicine, Yale University, New Haven, CT 06520, USA; alexei.surguchev@gmail.com

**Keywords:** BAP1, cancer, malignant melanoma, melatonin, neurotrophic factors, Parkinson’s disease, synucleins, transcription factors, ubiquitin–proteasome system

## Abstract

Epidemiological evidence points to an inverse association between Parkinson’s disease (PD) and almost all cancers except melanoma, for which this association is positive. The results of multiple studies have demonstrated that patients with PD are at reduced risk for the majority of neoplasms. Several potential biological explanations exist for the inverse relationship between cancer and PD. Recent results identified several PD-associated proteins and factors mediating cancer development and cancer-associated factors affecting PD. Accumulating data point to the role of genetic traits, members of the synuclein family, neurotrophic factors, the ubiquitin–proteasome system, circulating melatonin, and transcription factors as mediators. Here, we present recent data about shared pathogenetic factors and mediators that might be involved in the association between these two diseases. We discuss how these factors, individually or in combination, may be involved in pathology, serve as links between PD and cancer, and affect the prevalence of these disorders. Identification of these factors and investigation of their mechanisms of action would lead to the discovery of new targets for the treatment of both diseases.

## 1. Introduction

A growing number of studies demonstrate that patients with PD have a lower risk of developing most types of cancer compared to the control group [1]. A major exception to this rule is melanoma, the incidence of which positively correlates with PD [2,3]. Experimental results demonstrating that PD and cancer might have several identical or similar genes and signaling pathways give a basis for the hypothesis that people predisposed to PD may have some mechanisms protecting them from cancer development [4]. New findings have provided experimental evidence that the negative correlation between PD and cancer may be explained by the fact that these two seemingly divergent diseases are connected by molecule linkers, which provide possible targets for their treatment [4]. Both conditions, apparently, can involve the same set of genes. However, in affected tissues, their expression is inversely regulated: genes downregulated in PD are upregulated in cancer and vice versa. Several candidates involved in both diseases may be considered as links connecting their pathogenesis (Table 1).

It is well established that accumulation of α-synuclein in PD causes some of the typical signs of this disease, while in malignant melanoma, elevation of α-synuclein increases the proliferation of tumor cells [4,5]. Other factors that may play roles as mediators between cancer and PD pathogenesis are neurotrophic factors, members of the ubiquitin–proteasome system, circulating melatonin, transcription factors, and mutations or polymorphisms in genes involved in the pathogenesis of these diseases. Below, we discuss the data supporting the role of these factors in more detail.

**Table 1 ijms-25-03899-t001:** Examples of pathogenic factors implicated in cancer and PD.

Neoplasm	Parkinson’s Disease
**Neurotrophic factors**	
*GDNF*	
Proliferation factor involved in the development and migration of gliomas [6].	Protective effects on neurons, including an influence on differentiation and survival of dopaminergic neurons [6].
*BDNF*	
Upregulated in various types of cancers [7].	Downregulated in PD and other neurodegenerative diseases [7].
**MicroRNAs**	
*miR-148a*	
Potential tumor suppressor. Reduces gastric cancer metastasis [8].	miR-148a expression decreased in patients with PD [8].
*miR-9, miR-29, and miR-34*	
Tumor suppressors in human carcinogenesis [9].	Differentially expressed in PD [9].
**Synucleins**	
*α-Synuclein*	
Expressed in cancerous tumors [10]. Cooperates with mGluR5 and γ-synuclein [11]. Involved in autophagy, mitochondrial metabolism, and ROS generation [12]. Regulates pro- and antiapoptotic processes [13]. Exosome-delivered α-synuclein inhibits liver cancer. α-Synuclein amyloids may affect cancer pathogenesis [14,15].	Familial PD is associated with mutations in the α-synuclein gene. Accumulation of misfolded α-synuclein oligomers increases vulnerability of neurons to dopamine-induced cell death. Affects mitochondrial metabolism, autophagy, and the generation of ROS [12].
γ-*Synuclein*	
High levels in many types of cancer [16]. Compromises mitotic checkpoint controls. Causes multinucleation and accelerated cell growth. Stimulates invasion; promotes metastasis [16] and, phosphorylation of TGF-β-induced p38 MAPK [17].	Expression of γ-synuclein and mGluR5 is reduced by the PD-associated protein α-synuclein [11]. γ-Synuclein modulates synaptic vesicle binding of α-synuclein and reduces α-synuclein’s physiological activity at the neuronal synapse [18].
**Polypeptide Irisin**	
Ameliorates the course of cancer. Decreases the expression of cancer markers [19,20].	Reduces the motor deficits induced by α-synuclein preformed fibrils and decreases pathological α-synuclein by enhancing its endolysosomal degradation [21].
**PINK1** (PTEN-induced kinase 1)	
Protective effects for some types of cancer. In ovarian and breast cancer, PINK1 is a tumor suppressor [22].	A neuroprotective effect, prevention of apoptosis. Cell protection due to Bcl-XL phosphorylation [23].
**DJ-1**	
Stabilizes the antioxidant regulator Nrf2. Modulates antioxidant responses [24,25].	Modulates antioxidant responses and maintains stability of Nrf2, shielding cells against oxidative insults [24,25].

## 2. Inverse Association between Cancers and PD

Many epidemiological studies have indicated an inverse association between the risk of developing cancers and PD [1,26,27,28]. For example, Bajaj and coauthors conducted a meta-analysis of 29 studies, including 107,598 PD patients (2010) [1]. The combined analysis demonstrated that the diagnosis of PD was associated with an overall 27% reduced risk of all cancers and a 31% diminished risk after the omission of melanoma and other skin tumors. In another study, a meta-analysis showed a 17% decreased risk of cancer in PD patients [29].

Although this correlation has been found in most studies, some exceptions have also been described. For example, in several investigations, leukemia, stomach, and uterine cancers did not show a significant, undisputed, and clear inverse association. Moreover, a positive association with certain cancers, including skin, breast, and brain cancers, was described [28]. Therefore, a more standardized approach with better criteria for enrollment and a higher number of patients is required to draw an unbiased conclusion about this association.

## 3. Differences and Similarities in the Pathogenesis of Cancer and PD

To better understand the reason for these controversies, it may be helpful to compare the mechanisms of cancer and PD development, to reveal the differences and similarities in their pathogenic pathways, and to disclose the roles of the main players in these diseases. Cancer and neurodegenerative processes are two main leading causes of morbidity and mortality worldwide [1,4,27]. At first glance, their pathogenesis is based on two apparently opposite biological processes. Cancer is characterized by aberrant and uncontrolled cellular multiplication and proliferation, whereas neurodegenerative diseases are associated with the progressive loss of structure or function of neurons. However, there are also some analogies between carcinogenesis and neurodegenerative diseases that may explain the controversies described above. Indeed, both diseases are caused by abnormal regulation of cell survival, and some data point to the existence of overlapping pathways associated with these two disorders [30]. Other similarities include the following: (a) Age-specific incidence rises sharply with age. (b) Clinical signs emerge late in the course of these diseases. (c) Marked geographical and environmental variations abound. (d) Events or exposures much earlier in life may underlie the later expression of the disease. (e) Both cancer and PD develop due to the interaction of genes and environmental factors. (f) Both diseases display a significant imbalance in the ubiquitination/deubiquitination processes.

In order to find new medications against these two devastating diseases, it is important to identify the reasons for the inverse association between the risk of developing cancers and PD and to identify underlying mechanisms.

## 4. Processes, Mechanisms, Pathways, and Factors That Can Explain the Inverse Relationship between Cancer and PD

### 4.1. Processes Controlling Functions of Mitochondria and Endoplasmic Reticulum

Mitochondria are important for all cellular activities, and mutations in genes controlling these subcellular structures affect tumorigenic process and neurodegeneration. Cancer and PD share common mutations in several mitochondrial proteins, e.g., Parkin and PINK1. As shown by Kalyanaraman and coauthors, mitochondria-targeted substances both possess neuroprotective properties and inhibit tumor cell proliferation (2020) [31]. Polyphenolic compounds, antioxidants, and other substances ensure the maintenance of cellular energy essential for neuronal cell survival. In contrast, energy conservation inhibits the proliferation of cancer cells by depriving them of the source of energy needed for cancer cell growth. Mitochondria-targeted drugs contain a triphenylphosphonium (TPP+) group attached with alkyl chains linked to a naturally occurring molecule [31]. These findings propose a drug-repurposing strategy based on mitochondria-targeted drugs with low toxicity.

A region located between the mitochondria and endoplasmic reticulum, termed the mitochondria-associated membranes (MAMs), plays a role in Ca^2+^ homeostasis and lipid synthesis. This structure requires an optimal distance between the endoplasmic reticulum and mitochondria for optimal function. A diminished distance has been described in PD and during cancer treatment [32]. The authors assume that the mitochondrial permeability transition pore is a crucial cell death signaling structure indirectly regulated by the spatial characteristics of MAMs.

MAMs participate in many fundamental biological processes, including calcium and lipid homeostasis, autophagy, inflammation, and apoptosis. MAMs also play an essential role in maintaining normal cellular functions, and the disturbance of MAMs by mutations or chemical agents may cause neurodegeneration and cancer development [33].

Thus, the processes controlling the functions of mitochondria and MAMs play an important role in the association between PD and cancer.

### 4.2. Mutations in Genes Controlling Apoptosis

One mechanism that might explain the relationship between cancer and PD is associated with gene mutations controlling the efficiency of apoptosis. Mutations increasing the efficiency of apoptosis might raise the risk of PD and decrease the probability of cancer. Polymorphic variations in the genes possessing such effects may also play some role [2].

Examining published data may also explain the association between PD and melanoma. This analysis shows that the changes in several genes increase the risk of both PD and melanoma, i.e., α-synuclein, Parkin, and LRRK2 [3]. Pan et al. (2012) reported that increased levels of α-synuclein in PD patients’ melanocytes reduced tyrosine hydroxylase levels, thereby reducing melanin synthesis and increasing the risk of melanoma. Therefore, increased levels of α-synuclein might raise the risk of PD and melanoma [34].

Similar analysis identified some low-penetrance genes, e.g., the cytochrome p450 debrisoquine hydroxylase locus, glutathione S-transferase M1, and the vitamin D receptor, that raise the risk of both PD and melanoma. The association between PD and melanoma may also be explained by impaired autophagy in PD and melanoma.

In a recent study, Strader and West (2023) [35] found that blood monocytes harbor elevated concentrations of LRRK2 and react to both intracellular and extracellular aggregated α-synuclein with strong pro-inflammatory responses. In addition to LRRK2, several other genes, e.g., α-synuclein, PTEN-induced kinase 1 (PINK1), F-box protein 7 (FBXO7), and ubiquitin C-terminal hydrolase L1 (UCHL1) show altered expression in cancer patients [36].

LRRK2 is involved in various cellular processes including inflammation, autophagy, cell survival, homeostasis, protein degradation, and mitochondrial functions [37].

Understanding the potential roles of these PD-associated genes and their gene products in lung cancer development can be beneficial in planning new therapeutic options for lung cancer. For instance, gene therapy can be exploited to regulate the expression profiles of these genes.

Additionally, advancement in CRISPR/Cas9 genome-editing technology is now being tested in clinical trials for lung cancer treatment [38]. In addition, gene-editing technology is being explored to combat drug resistance in various cancers such as breast, colon, and prostate cancers [39,40,41].

Gong et al. (2017) [42] showed that mutations in the PARK2 gene can promote both PD and cancer. One of the mechanisms involved is the ability of PARK2 tumor suppressors to control the apoptotic regulator Bcl-XL and modulate programmed cell death. PARK2 directly binds to and ubiquitinates Bcl-XL. Inactivation of PARK2 causes aberrant accumulation of Bcl-XL both in vivo and in vitro. On the other hand, cancer-specific mutations in PARK2 abolish the ability of the ubiquitin E3 ligase to target Bcl-XL for degradation. Moreover, PARK2 regulates mitochondrial depolarization and apoptosis in a Bcl-XL-dependent manner. Thus, the PARK2 tumor suppressor is able to exert its antiproliferative action by regulating cell cycle progression and programmed cell death.

Therefore, mutations in several genes controlling apoptosis may affect the development of both cancer and PD, serving as the link between these two disorders.

### 4.3. Ubiquitin–Proteasome System

A common pathway in PD and cancer is the ubiquitin–proteasome system (UPS), which controls protein degradation and cell cycle [43]. The UPS is an essential regulatory structure in cells that supports cellular homeostasis, controls signaling transduction, and affects cell fates. The study of mutations in the gene encoding E3 ubiquitin ligase (PARK2) has contributed to our understanding of the connection between PD and cancer. Changes in PARK2 affecting E3 ubiquitin ligase activity are the most common cause of early-onset PD. Veeriah and coauthors [44] demonstrated that the PARK2 mutations in cancer occur in the same domains as the germline mutations causing familial PD. Cancer-specific mutations abolish the growth-suppressive effects of the PARK2 protein.

Furthermore, PARK2 mutations in cancer reduce PARK2′s E3 ligase activity, compromising its ability to ubiquitinate cyclin E and resulting in mitotic instability. These findings point to PARK2 as a tumor suppressor. PARK2 is a gene that causes neuronal dysfunction when mutated in the germline, and it may also contribute to oncogenesis when altered in non-neuronal somatic cells [44].

Disturbances in this system are described in cancer, where the system is overactive [43,45,46], and in PD, where the UPS pathway is impaired [47,48,49,50]. UPS inhibitors are considered low-invasive chemotherapy drugs and are progressively used to relieve symptoms of various cancers in malignant states. There are two ways to synthesize a scaffold of UPS inhibitors and change it. The first approach uses the biology-oriented synthetic method to produce structurally novel molecules. The second approach is fragment-based compound design, in which several fragments are used to generate a scaffold by conjugating with each other [43].

Tian et al. (2021) investigated the genetic linkage between PD and gastric cancer using the barcode algorithm [51]. The method makes it possible to analyze overlapping differentially expressed genes in two diseases. This approach revealed that the ubiquitin-conjugated enzyme E2 M protein (UBE2M) is linked to both diseases. In addition to UBE2M, three other candidates were identified: cathepsin D (CTSD), glutathione peroxidase 3 (GPX3), and casein kinase 1 delta (CSNK1D). Further analysis will show the exact mechanism of their involvement.

Another potential link between PD and cancer is ubiquitin C-terminal hydrolase BRCA1-associated protein 1 (BAP1), a tumor suppressor and a known genetic risk factor for PD [52,53]. BAP1 is a ubiquitin C-terminal hydrolase with a wide array of biological activities.

These results point to the ubiquitin–proteasome system, a common pathway in PD and cancer that controls protein degradation, as a potential linking mechanism between these two diseases.

### 4.4. Circulating Melatonin

Accumulating data point to the role of circulating melatonin in the association of PD with cancer. Schernhammer et al. [54] assumed that elevated circulating melatonin levels in patients with PD cause reduced cancer risk. There are significant laboratory data and some epidemiological evidence suggesting that melatonin is important in carcinogenesis. The findings of elevated morning melatonin levels in patients with PD compared to healthy controls partially support this hypothesis. However, the evidence about melatonin’s role in PD and other neurodegenerative diseases is not conclusive and needs to be further investigated.

Hardeland (2012) [55] found that a substantial reduction of circulating melatonin occurred in neurological disorders and cancer.

The interplay between circadian oscillators and melatonin secretion holds a significant role in cellular homeostasis. Importantly, the readjustment of rhythms controlled by melatonin and its synthetic analogs may be used to improve the course of circadian-rhythm-dependent disorders.

### 4.5. Transcription Factors

A growing number of experimental results show that transcription factors (TFs) are involved in the modulation of age-related disorders, including cancer and neurodegenerative diseases. They play a crucial role as regulators of many important cellular processes, including mitochondrial biogenesis, energy metabolism, oxidative stress, DNA repair, inflammation, nutrient homeostasis, vascular development, and cell regenerative capacity [56]. For example, transcription factor EB (TFEB) participates in the regulation of DNA damage and epigenetic modifications, inducing autophagy and proteostasis, regulating mitochondrial quality control, and linking nutrient sensing to energy metabolism. TFEB also modulates pro- and anti-inflammatory pathways, suppresses senescence, and stimulates cell regeneration. Safe and effective strategies for activating TFEB might be considered as a therapeutic approach for cancer and neurodegenerative disease treatment and for extending longevity [56].

Nabar et al. (2021) [57] demonstrated that LRRK2 is an upstream activator of TFEB, which is a host defense transcription factor and the master transcriptional regulator of the autophagy/lysosome machinery. LRRK2 mutation stimulates hyperactivation of TFEB, both by stabilizing TFEB and by promoting its nuclear translocation via aberrant calcium signaling.

In Moor et al.’s study (2017) [58], the authors pointed to an abnormal regulation of autophagy associated with the aggregation of α-synuclein. Since autophagy is the pivotal system in the proteolytic degradation of α-synuclein, its pharmacological enhancement may be an attractive strategy to combat α-synuclein aggregation. Another important finding is that TFEB should be considered as a pharmacological target, since this transcription factor may be used as an autophagy regulator due to its specific effect on molecular pathogenetic processes causing PD and cancer.

Another TF playing a key role in cancer and neurodegeneration is HIF1A, a hypoxia-inducible factor considered the master transcriptional regulator of cellular and developmental responses to hypoxia [59]. This TF responds to reductions in available oxygen in the cellular environment and can be modified by E3 ubiquitin ligase (Parkin). Liu et al. demonstrated that HIF1A ubiquitinated by Parkin (E3 ubiquitin ligase) on lysine 477 (K^477^) became susceptible to degradation, which, in turn, reduced metastasis of breast cancer cells [59]. On the other hand, Parkin plays a critical role in mitochondrial quality control, and mutations in the Parkin gene cause a form of autosomal recessive juvenile PD [60].

There are other proteins with TF activity that are implicated in both cancer pathogenesis and neurodegenerative diseases, which may play a role in linking their pathogenic pathways. For example, DJ-1, a cancer- and PD-associated protein, stabilizes the antioxidant transcriptional master regulator Nrf2 (nuclear factor erythroid 2-related factor), a master regulator of antioxidant transcriptional responses [24,25]. DJ-1 stabilizes Nrf2 by preventing association with its inhibitor protein, Keap1; this causes Nrf2′s subsequent ubiquitination. Without intact DJ-1, Nrf2 protein is unstable, and both its basal and induced transcriptional responses are decreased. DJ-1′s effect on Nrf2 and following influence on antioxidant responses can explain how DJ-1 affects both cancer and PD. The DJ-1/Nrf2 functional axis represents a therapeutic target for cancer treatment and confirms the role of DJ-1 as a tumor biomarker [24,25].

Thus, transcription factors, critical regulators of many cellular processes, play a role as links between PD and cancer. 

### 4.6. Amyloidogenic Substances

Naskar and Goar analyzed published data and proposed a generic amyloid hypothesis considering that amyloidogenic proteins may be responsible for the etiology of a plethora of diseases including PD and cancer [61]. The hypothesis assumes that traditional amyloids are formed not only by proteins or peptides but also by metabolites, including single amino acids, nucleobases, lipids, and glucose derivatives. They all have a propensity to form amyloid-like toxic assemblies. For example, phenylalanine can generate amyloid-like nanofibrillar structures at millimolar concentrations [62]. The authors assume that common therapeutic interventions for these diseases can be developed by designing drugs that act as generic amyloid inhibitors.

### 4.7. Neurotrophic Factors

Neurotrophic factors possess protective effects on neurons, including their influence on the differentiation and survival of dopaminergic neurons. For example, glial cell line-derived neurotrophic factor (GDNF) is a member of the transforming growth factor β (TGF-β) superfamily. GDNF plays the role of a survival factor, acting on different neuronal activities, and is now an established therapeutic target in PD [6].

On the other hand, accumulating evidence demonstrates that GDNF is abundantly expressed in gliomas, especially in glioblastomas, and is a potent proliferation factor involved in the development and migration of gliomas. For example, it is highly expressed in glioblastoma multiforme (GBM) due to the epigenetic regulation of its gene expression. This epigenetic mechanism includes DNA methylation and histone H3K9 acetylation of the promoter and silencer regions of the GDNF gene, stimulating its transcription. GDNF also modulates microtubule-associated proteins and actin-associated proteins in cellular migration. Due to these activities, GDNF may acquire pro-oncogenic activity and deprive cells of their apoptotic activity. As a result of high GDNF activity, microglia are attracted to the tumor microenvironment to promote glioma progression. Thus, GDNF may contribute to glioma migration and invasion [6] and play the role of a potential molecular link in the association between PD and glioma.

The expression of another neurotrophic factor, brain-derived neurotrophic factor (BDNF) is downregulated in PD and other neurodegenerative disorders and upregulated in various types of cancers. A lower level of BDNF in PD is related to cognitive and other neuropsychological deficiencies. At the same time, high concentrations of BDNF are associated with tumor growth, metastasis, and poor survival in cancer patients [7]. The authors assume that the BDNF level is essential in establishing the program of cellular pathophysiology, being a critical factor in regulating homeostasis and the development of cancer or neurodegenerative disorders.

As shown by Jiang et al. (2018) [63] the blood levels of BDNF are decreased in PD patients. The reduction in BDNF’s blood levels might be due to the chronic inflammatory state of the brain in neurodegenerative disorders, since neuroinflammation affects several BDNF-related signaling pathways [7].

Currently, a growing number of findings confirm the leading role of neurotrophic factors in the association between PD and cancer.

### 4.8. Chronic Inflammation

Another link between cancer and PD is chronic inflammation in neurons and tumors, which contributes to microenvironmental changes that cause the accumulation of DNA mutations and facilitate disease development [64]. This hypothesis is based on consideration of the crucial role of microglia and the genetic involvement of COX2 and CARD15 in PD and cancer. If the main findings of this article are confirmed, it will explore preventive and therapeutic measures for both disorders [64].

Inflammation is a part of the innate immune response following insults to the body. Inflammation within the CNS is part of the pathogenesis of neurodegenerative diseases, particularly PD [64].

### 4.9. MicroRNAs

MicroRNAs (miRNAs) are short ncRNAs crucial for gene expression regulation through binding to target mRNAs. These molecules typically consist of 20 to 22 nucleotides and control various biological processes, emerging as potential biomarkers in different pathological conditions.

MiRNAs may play a role as a link between PD and cancer. Saito and Saito reported that miR-9, miR-29, and miR-34 are differentially expressed in PD and other neurodegenerative diseases and act as tumor suppressors during human carcinogenesis [9]. Hu and coauthors found that miR-148a is a potential tumor suppressor that reduces gastric cancer metastasis and is involved in neurological development and several other processes [8]. For example, the expression of miR-148a is decreased in patients with PD compared to that in the control group.

It is evident that the set of miRNAs identified as modulators of both neurodegeneration and cancer development is currently incomplete, and the list will continue to grow.

## 5. α-Synuclein at the Crossroads of PD and Cancer

Synucleins are a family of three small, conserved proteins (α-, β-, and γ-synucleins) expressed primarily in neural tissue and some tumors [65]. Despite the lack of clear understanding of their physiological role, their association with neurodegenerative diseases and cancer has been attracting the attention of researchers for many years.

### 5.1. α-Synuclein in Synucleinopathies

It is well established that α-synuclein plays a fundamental role in PD and other synucleinopathies. Accumulation of misfolded α-synuclein oligomers and larger aggregates raises the vulnerability of neurons to dopamine-induced cell death [66,67,68]. Furthermore, α-synuclein is the major protein component of Lewy bodies and Lewy neurites [68,69]. The association of α-synuclein with PD and other synucleinopathies is proven by various experimental approaches, including genetic, biochemical, and immunochemical, and is summarized in several reviews [67,68,69]; therefore, we will not discuss these details here.

### 5.2. Synucleins in Cancer

The association of synucleins with cancer was first documented for γ-synuclein in breast cancer [70] and later in several other types of cancer [43,71,72,73]. Accumulating data indicates that two other members of the synuclein family (α and β) are also associated with tumorigenesis (reviewed in [12]]. For example, both β-synuclein and γ-synuclein are highly expressed in stage III-IV of breast ductal carcinomas, while all three types of synucleins are present in ovarian carcinomas [72].

α-Synuclein is expressed in certain types of cancer, i.e., its expression is described in breast cancer, melanoma, and ovarian cancer [12,72,74]. The presence of α-synuclein was also reported in medulloblastomas [75], colorectal cancer [73], acute erythroid leukemia, acute megakaryoblastic leukemia [76], and osteosarcoma cells [77]. According to Kawashima et al., α-synuclein is expressed in various brain tumors showing neuronal differentiation (2000) [10]. The data about the changes in α-synuclein expression level depends on the type of cancer. An increased expression of α-synuclein is described in pancreatic adenocarcinoma [78] and melanoma [79]. On the other hand, α-synuclein expression is reduced in some tumor tissues, and its downregulation is related to poor prognosis and overall survival [80]. There is a substantial decrease in α-synuclein expression in colon adenocarcinoma; however, in colorectal cancer, the α-synuclein level is increased [12]. In some studies, it was demonstrated that upregulation of α-synuclein suppresses tumorigenesis [43].

New results recently published by Yang suggest a mechanism of α-synuclein involvement in cancer development. α-Synuclein acquires this activity after combining with two other proteins implicated in tumorigenesis: metabotropic glutamate receptor 5 (mGluR5) and another member of the synuclein family, γ-synuclein (2023) [43] (Table 1) [11]. mGluR5 receptor activity is essential for the proliferation and survival of cancer cells [81]. These new findings demonstrate that mGluR5 and γ-synuclein are tumor-promoting factors stimulating cancer cells migration, proliferation, and metastasis. Thus, α-synuclein can be associated with mGluR5 and γ-synuclein, contributing to the inhibitory effect on cancer progression [43]. The involvement of α-synuclein in various types of cancer is summarized in a recent review by Zanotti et al. [12].

Synuclein’s role in cancer is associated with its involvement in various cancer-related cell signaling processes and pathways. For example, α-synuclein is implicated in autophagy, mitochondrial metabolism, and the generation of ROS in cancer and the same processes in neurodegenerative diseases [12].

### 5.3. α-Synuclein and Melanoma

The association between PD and malignant melanoma was found not only in epidemiological studies, as discussed above (Section 2), but also in biochemical studies. This relationship is due, at least partially, to α-synuclein’s role as a regulator of pro- and antiapoptotic processes [13]. The accumulation of α-synuclein oligomers has been described as a hallmark of PD. It is also observed in malignant melanoma, where it contributes to the increased proliferation of tumor cells [5,82,83,84].

Furthermore, in cultured B16 melanoma cells, α-synuclein overexpression causes increased cell proliferation [85]. Importantly, knocking out α-synuclein in melanoma cells suppresses tumor growth [86]. Accumulating data point to the epigenetic mechanisms, which play a critical role regulating α-synuclein expression and affect its involvement as a mediator linking cancer and neurodegeneration [4,87,88].

The data showing that α-synuclein expression influences tumorigenesis [85,89] point to the existence of a common pathogenic mechanism(s) between cancer and neurodegenerative diseases. Additional evidence in favor of such a mechanism is the effect of a higher risk of malignant melanoma in PD patients [90,91,92].

Filippou and Outeiro [93] speculated that comprehension of both PD and cancer may have synergetic effects in understanding the mechanism of pathology and help identify new biomarkers and targets for intervention, hopefully leading to improved diagnosis and management of both diseases.

According to the majority of data, the inverse correlation between α-synuclein expression and melanin production suggests that α-synuclein disturbs the activities of tyrosine hydroxylase and tyrosinase—enzymes involved in melanin biosynthesis [2,4]. Dean and coauthors proposed an alternative explanation of the interplay between α-synuclein and melanin (2021) [94]. Based on epidemiological findings and molecular interaction data, they assumed that α-synuclein binds and controls the aggregation of Pmel17, a functional amyloid that serves as a scaffold for melanin synthesis.

Thus, the disruption of melanin biosynthesis by α-synuclein in melanoma cells may involve two amyloid-forming proteins, α-synuclein and Pmel17, affecting the pigmentation in melanoma cells [94].

Lee et al. (2013) found physiological significance of α-synuclein phosphorylation at serine-129 [95]. These researchers demonstrated that such phosphorylation causes α-synuclein translocation to the cell surface, with its subsequent vesicular release in melanoma cells. The authors hypothesized that α-synuclein release from melanoma cells plays a role in the pathogenesis or progression of PD, since it might propagate into neuronal cells. This hypothesis is partially supported by the finding that α-synuclein is transmitted from human melanoma cells to neuroblastoma cells when these two types of cells are co-cultured [96]. Interestingly, Ser129-phosphorylated α-synuclein is abundantly found in melanoma cells but not in normal skin [2].

#### Role of α-Synuclein Phosphorylation

α-Synuclein post-translational modifications change its properties and affect its involvement in diseases, including PD and cancer. Currently, the most thoroughly studied post-translationally modified α-synuclein is the form that is phosphorylated at serine-129 (Ser^129^) [97]. Almost all α-synuclein in Lewy bodies in PD brains is phosphorylated at Ser^129^ [98,99,100,101]. Phosphorylation of α-synuclein at Ser^129^ also affects its role in melanoma. According to Inzelberg and coauthors, the mechanisms underlying the high prevalence of cutaneous malignant melanoma in PD involve common pathways in which α-synuclein phosphorylated on Ser^129^ plays a major role. These authors found that this form of α-synuclein could modulate Pmel17 functional amyloid formation [2]. This post-translationally modified form of α-synuclein is abundantly expressed in cutaneous malignant melanoma but not in normal skin [83].

## 6. Molecular Mechanisms Underlying α-Synuclein’s Role in the Association between PD and Cancer

Hou et al. (2023) examined the potential role of α-synuclein in the link between PD and liver cancer. They found that exosome-delivered α-synuclein inhibited the growth, migration, and invasion of cultured hepatocellular carcinoma cells [30]. Integrin αVβ5 in exosomes enhanced this effect. Importantly, in vivo experiments with rat models confirmed that exosome-delivered α-synuclein inhibited liver cancer. These findings show the important role of α-synuclein in the inhibition of hepatoma, suggesting a new mechanism underlying the link between these two diseases.

Recent findings suggest the role of epigenetic mechanisms, which play a critical role in the regulation of α-synuclein expression and affect its involvement as a mediator linking cancer and neurodegeneration [4,87,88].

### 6.1. Chemical Reactivity of α-Synuclein Amyloids

Recent studies point to a new gain-of-function modification of α-synuclein amyloids that may affect the pathogenesis of human diseases. α-Synuclein amyloid fibers possess enzyme-like catalytic properties, for example, esterase and phosphatase activity, whereas α-synuclein monomers have little or no enzymatic activity [14]. In another recent article, Horvath and Wittung-Stafshede demonstrated that α-synuclein amyloid caused alterations in metabolites: the amounts of four of them increased, while the amounts of seventeen decreased [15] (Figure 1).

The exact nature of these findings deserves further investigation, since they may open a new era dealing with unexplored pathological pathways associated with amyloids in human diseases.

One of the possible consequences of α-synuclein amyloid accumulation is an influence on the pathogenesis of other human diseases. This is plausible because α-synuclein amyloids acquire deleterious gain-of-function effects and might cause mitochondrial dysfunction, oxidative stress, and protein degradation failure [14,15,102].

### 6.2. Irisin Is at the Crossroads between Cancer and PD

Irisin is an adipomyokine that is involved in the regulation of metabolic processes. It is an exercise-induced polypeptide secreted by skeletal muscle. Irisin is at the crossroads between cancer and PD, being able to ameliorate the course of both diseases. It has been confirmed that irisin inhibits in vitro proliferation, migration, and invasion. It also influences inflammatory processes, including cancer, and decreases the expression of cancer markers [19,20].

On the other hand, irisin prevents pathological α-synuclein-induced neurodegeneration in PD. It causes a decrease in the formation of pathological α-synuclein, prevents the loss of dopamine neurons, and lowers striatal dopamine. Irisin also reduces motor deficits and decreases the formation of phosphorylated Ser^129^ in α-synuclein [21].

## 7. Roles of Two Other Members of the Synuclein Family, β- and γ-Synuclein, in Pathology

### 7.1. β-Synuclein

β-Synuclein is closely related to α-synuclein [65], and these two members of the family are often coexpressed [103]. β-Synuclein acts as a molecular chaperone to inhibit α-synuclein aggregation [104]. Recent results show that β-synuclein is involved in synergistic and antagonistic interactions with α-synuclein and participates in cellular pathways independent on α-synuclein [105,106,107,108]. Moreover, β-synuclein promotes neurotoxicity, indicating that β-synuclein is implicated in other cellular pathways independent of α-synuclein. β-Synuclein is involved in neurodegenerative diseases [105], and P^123^H and V^70^M mutations in β-synuclein are associated with dementia with Lewy bodies [104]. β-Synuclein also attracts attention in the neuro-oncological area owing to its high expression in glioma tissues [106] in erythroid leukemia and acute megakaryoblastic leukemia [107]. β-Synuclein is implicated in age-related as well as pathophysiological conditions and regulates p53-mediated and Akt-independent apoptosis [109]. Accumulating new data point to an emerging concept of an α-synuclein-independent pathway of β-synuclein [110,111].

### 7.2. γ-Synuclein

γ-Synuclein is a predominantly neuronal protein that is also overexpressed in various types of human cancer. High levels of γ-synuclein protein have been revealed in many types of cancer, notably in the advanced stages of the disease [16]. Furthermore, overexpression of γ-synuclein compromises normal mitotic checkpoint controls, causing multinucleation and accelerated cell growth. γ-Synuclein stimulates invasion and promotes metastasis in in vitro assays and animal models. In addition, γ-synuclein promotes phosphorylation of transforming growth factor-β-induced p38 mitogen-activated protein kinase (MAPK) [18]. These findings point to the essential role of p38MAPK in promoting cancer metastasis via γ-synuclein, implying that p38MAPK inhibitors may serve as potential medications for γ-synuclein-overexpressing cancer [18].

Motor neurons contain a significant amount of γ-synuclein especially in axons, where it presumably regulates the organization of the axonal cytoskeleton [112]. Accumulation of γ-synuclein in distinct profiles within the dorsolateral column has been described in amyotrophic lateral sclerosis (ALS) cases [113]. Histopathological structures containing abnormal γ-synuclein have been reported in several neurodegenerative diseases [114,115,116,117]. Although γ-synuclein’s role in several types of cancer and neurodegenerative diseases is well documented, there are no clear data about its role in the association of these two types of pathologies.

Synucleins’ role in neurodegeneration is confirmed by many experiments, while their involvement in cancer is a relatively new area of investigation. However, the data about their association with various cancer-related cell signaling processes and pathways, bringing a new understanding of their role, deserve further attention.

## 8. Common Protective Factors against PD and Cancer

Reduced levels of cancer mortality or incidence in PD patients have generated an assumption about the existence of protective factors common to both diseases. It is important to identify these factors to understand the mechanisms of both pathologies better and to find approaches for their treatment. One factor that may play a protective role in both PD and cancer is the serine–threonine mitochondrial protein kinase PINK1. PINK1 possesses a neuroprotective effect, since it prevents mitochondrial damage and apoptosis in response to stress factors. Cell protection by PINK1 is due to Bcl-xL phosphorylation and its pro-apoptotic cleavage regulation [22]

On the other hand, PINK1 has some protective effects against some types of cancer, and its loss results in elevated proliferation of glioma cells, decreased oxygen consumption, and raised glycolysis [23]. In certain types of malignancy, e.g., ovarian and breast cancer, PINK1 plays the role of a tumor suppressor [22].

Another protein that may protect against cancer and PD is Parkin (Park2), a multifaceted E3 ubiquitin ligase. It is an important tumor suppressor, and at the same time, it inhibits apoptosis and promotes survival in neuronal cells, potentially being protective against cancer and PD. Such protective activity of Parkin may be associated with its regulatory action toward p53. Inhibition of p53-mediated apoptosis is a mechanism that causes the neuroprotective effect of Parkin [118].

## 9. Conclusions

Multiple clinical and epidemiological studies have shown that patients with PD have a lower risk of developing cancer. However, the mechanisms of their association still need to be completely understood. PD and cancer are intricate diseases with multiple cellular changes. An intriguing aspect of the inverse association between PD and cancer that merits thorough investigation is their different pathophysiological time frames. PD is characterized by gradual neuronal loss, being a chronic and generally slowly progressing neurodegenerative disease.

In contrast, cancer usually exhibits rapid progression, with fast proliferation of glial cells over a much shorter duration. This disparity suggests that in PD, neuronal loss can be compensated for over an extended time, whereas the aggressive nature of cancer, driven by highly infiltrative and metastasizing cells displaying considerable heterogeneity, leads to a rapid disease progression. Common pathogenic mechanisms underlie both PD and cancer, encompassing inversely deregulated pro-survival and immune signaling, DNA damage, mitochondrial dysfunction, cell cycle defects, metabolic alterations, and chronic inflammation. Recent data points to neurotrophic factors and members of the synuclein family as signaling molecules involved in both diseases.

Recent accumulating data highlights new mechanisms and pathways linking Parkinson’s disease (PD) and cancer. For instance, pollutant-induced ferroptosis has emerged as a process connecting PD and cancer [119]. Ferroptosis is a form of programmed cell death that relies on iron and is characterized by the accumulation of lipid peroxides induced by pollutants. It is genetically and biochemically distinct from other regulated cell death mechanisms such as apoptosis [119]. Ferroptosis is associated with an imbalance in iron homeostasis, leading to excessive intracellular iron deposition in brain cells. This disruption in iron balance adversely affects the brain, contributing to the development of PD, cancer, and other human diseases [120].

Furthermore, recent findings have revealed the involvement of ncRNAs in both PD and cancer development. Shou et al. (2024) [121] demonstrated that long ncRNAs, such as ANRIL, are implicated in both malignant processes and PD by influencing gene expression modulation, splicing events, and mRNA stability.

## 10. Future Directions

Advancements in understanding the pathogenesis of PD and cancer have the potential to catalyze the development of novel diagnostic methods with practical clinical applications. Insight into the potential mechanisms underlying these pathologies and their interdependencies could be the foundation for effective targeted therapies, modifying and enhancing the course and prognosis of both diseases. Increasing knowledge about the involvement of common pathways, mechanisms, and molecules in these two diseases will aid in devising strategies to combat the challenges on both fronts and decipher the molecular connections between PD and cancer.

Advancing our understanding of the common pathways, mechanisms, and molecules involved in these two diseases will help devise strategies to address challenges on both fronts and unravel the molecular connections between PD and cancer. The rapid progress in CRISPR technology holds promise for treating genetic forms of these diseases, while the development of artificial intelligence will uncover existing associations and identify new ones.

Although CRISPR/Cas9 genome-editing technology is in an early stage of development, there are examples of its successful application. For example, this technique is being tested in clinical trials for lung cancer treatment [38]; it is also being explored to fight drug resistance in different cancers, including colon, breast, and prostate cancers [39,40,41].

Artificial intelligence (AI) can be used to analyze proteomic data to identify patterns indicative of early stages of disease [122]. Proximity-labeling-based methods coupled with mass spectrometry offer a high-throughput approach for analyzing spatially restricted proteomes. Proximity labeling exploits enzymes that generate reactive radicals to tag neighboring proteins covalently. LaPak et al. [123] used this method to identify binding partners of NRF2 and related family members. They found that the transcription finger ZNF746 (PARIS), which is increased in PD, binds to and represses the activity of NRF2.

Artificial intelligence (AI) is being successfully used in nuclear medicine, focusing on several fields: neurology, oncology, and cardiology. AI is efficient for diagnosis and treatment planning for patients with PD, Alzheimer’s disease, and cancer [124]. AI-based prediction models are being developed, playing an increasing role in screening, diagnosis, treatment selection, and rescue therapy decision making. AI also helps in image interpretation, putting it at the forefront of the next phase of personalized medicine.

Machine learning systems and radiomic features have been successfully used for the prediction of pathogenic variants of PD [125].

## Figures and Tables

**Figure 1 ijms-25-03899-f001:**
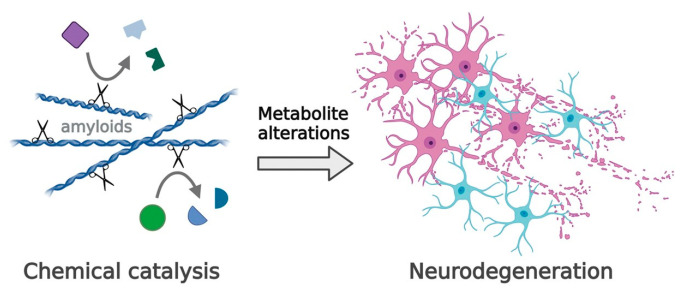
Chemical reactivity of pathological amyloids may be an unexplored gain-of-function in amy-loid-related diseases (such as neurodegenerative disorders). Amyloid-mediated chemicals may promote alterations in metabolite composition in cells and thereby modulate disease progression (e.g., accelerate neurodegeneration). Created with BioRender.com. Reproduced from Wittung-Stafshede P. Chemical catalysis by biological amyloids. *Biochem Soc Trans.* 2023; 51(5):1967–1974 [102].

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
