# Peer review of "Association between Parkinson’s Disease and Cancer: New Findings and Possible Mediators"

_ijms, 2024, doi:10.3390/ijms25073899_

Round 1
Reviewer 1 Report
Comments and Suggestions for Authors
Author Response
Responses to Reviewer 1.
Thank you very much for your valuable critiques and comments which allowed us to improve our manuscripts. Below are point-by-point answers.
1 A more organized format for Table 1 including space and word size would be highly recommended.
We reformatted Table 1 and included its updated version in the manuscript.
2 Lines 111-115 Recent studies emphasizing the correlation of cancer and PD. Particularly, studies in genetics, for example, LRRK2
We added the following text and 2 references 35 and 36 in response:
In a recent study Strader and West (2023) found that blood monocytes harbor elevated concentrations of the LRRK2 and respond to both intracellular and extracellular aggregated α-synuclein with strong pro-inflammatory responses. In addition to LRRK2, several other genes, e.g. α- synuclein, PTEN-induced kinase 1 (PINK1), F-box protein 7 (FBXO7) and ubiquitin C-terminal hydrolase L1 (UCHL1) alter expression in cancer patients (Leong et al., 2023)
3 Lines 171-180 Is there any research focusing on the risk of carious …
We failed to find any reliable research data on the risk of carious playing a role in the association of PD with cancer.
4 Lines 221-3 Clarify whether the reduced BDNF level in PD is in brain autopsied tissues, CSF or cell base experiments.
We added the following text and a reference in response:
As shown by Jiang et al. (2018) the blood levels of BDNF are decreased in PD patients.
63 Jiang L, Zhang H, Wang C, Ming F, Shi X, Yang M. Serum level of brain-derived neurotrophic factor in Parkinson's disease: a meta-analysis. Prog Neuropsychopharmacol Biol Psychiatry. 2019, 10; 88:168-174. doi: 10.1016/j.pnpbp.2018.07.010.
5 lines 229-235 Add chronic inflammation in cancer and PD. Role active and passive immune system
We added texts dealing with inflammation in the corresponding section; we have it in sections 4.7, 4.8, 9 and in references 25 and 64.:
6 Lines 236- 237 Introduce background knowledge miRNAs
We added the following introduction for microRNAs (lines 301-304):
MicroRNAs (miRNAs) are short non-coding RNAs crucial for gene expression regulation through binding to target mRNAs. These molecules typically consist of 20 to 22 nucleotides and participate in the control of various biological processes, emerging as potential biomarkers in different pathological conditions.
7 Line 249 Subtitle - typo. Typo is corrected
We corrected α-. Synuclein as α-Synuclein (line 317)
8 Improve grammar and structure for lines 264-267, 280-289, 306-320, 318-320,342-344, 388-390, 412-415, 420-426. The grammar and structure are corrected in all texts mentioned by the reviewer.
9 Lines 360-365 the correlation of chemical activity in aSyn amyloid structure in cancer development is elusively interpreted.
We added the following text (lines 445-448):
One of the possible consequences of α-synuclein amyloid accumulation is the influ-ence on the pathogenesis of other human diseases. This is plausible because α-synuclein amyloids acquire deleterious gain-of-functions effects and might cause mitochondrial dysfunction, oxidative stress, and protein degradation failure [14, 15, 102].
10 Lines 452-457: Provide related studies to explain how we can utilize CRISPe and AI technologies to study interplay between PD and cancer.
In response to this comment, we added the text on lines 546-564 and corresponding references.
The text begins:
“Although CRISPR/Cas9 genome editing technology is on an early step of development.. “
Reviewer 2 Report
Comments and Suggestions for Authors
This manuscript reviews the association between Parkinson’s disease (PD) and cancer. Although PD and cancer have been extensively reviewed in recent years, the authors have a unique perspective in focusing on the relationship of them. Overall, the manuscript is well written and organized. Some minor points should be addressed before the manuscript is ready for publication.
1. It would be helpful if the abbreviations list is included.
2. It would be better if Table 1 is on a single page.
3. PARK2 is also named as Parkin, and PARK6 is known as PINK1. Since the authors use “PARK2” in the main text while “PINK1” in Table 1, it is better to annotate the names when they first appeared.
Comments on the Quality of English LanguageThe writing is logical and clear. Language is good.
Author Response
- It would be helpful if the abbreviations list is included.
Abbreviations list is added, thank you.
- It would be better if Table 1 is on a single page.
We reformatted Table 1, thanks
- PARK2 is also named as Parkin, and PARK6 is known as PINK1. Since the authors use “PARK2” in
the main text while “PINK1” in Table 1, it is better to annotate the names when they first appeared.
Thank you. We made annotations for E3 ubiquitin ligase (PARK2) in Section 4.3 and for PINK1 in Table 1.
Reviewer 3 Report
Comments and Suggestions for Authors
The manuscript “Association of Parkinson's disease and cancer: new findings and possible mediators” offers a captivating exploration of the association between Parkinson's disease (PD) and most cancers.
The authors effectively guide readers through current epidemiological evidence and potential biological mechanisms, highlighting shared pathogenetic factors and mediators that hold promise for novel therapeutic strategies. However, certain areas could be refined to further strengthen the manuscript's clarity, and overall impact.
Within the main manuscript, I encourage the authors to address concerns regarding the flow and coherence of section 4.1. Ensuring smooth transitions between sentences and ideas will greatly facilitate comprehension. Furthermore, section 4.2 would benefit from a more detailed exploration of the potential role of apoptosis-promoting mutations, accompanied by comprehensive and relevant references to support the statements. Similarly, comprehensive references are essential to substantiate claims in section 4.4, unless solely based on Schernhammer and coauthors' work.
I commend the authors for recognizing the intriguing potential of shared pathogenetic factors and mediators. Two areas, in particular, present fertile ground for further exploration: transcription factors and chronic inflammation.
Section 4.5 highlights the importance of transcription factors, particularly TFEB. As the authors acknowledge, TFEB activation has been shown to stimulate cellular regeneration, offering a glimmer of hope for neurodegenerative diseases like PD. However, its potential relevance to cancer therapy required a more detail explanation. I recommend delving deeper into its potential relevance for cancer therapy by exploring its interplay with cancer-suppressing pathways, such as autophagy, providing specific examples and pertinent references.
Chronic inflammation, while mentioned in section 4.8, warrants a more comprehensive examination. Expanding on this theme by providing a more comprehensive characterization can offer valuable insights into their shared pathogenic mechanisms and pave the way for novel therapeutic strategies. To deepen the understanding of chronic inflammation's role, the authors could consider outlining its distinct features in both disease. This detailed characterization could encompass: Cellular players (Identifying the specific immune cells and inflammatory mediators involved in each disease, highlighting potential overlaps and synergies) and Signaling pathways (Exploring the intricate signal transduction pathways responsible for sustained inflammation). Additionally, exploring the intricate signal transduction pathways responsible for sustained inflammation, with a focus on potential points of convergence for therapeutic intervention, would significantly strengthen the manuscript's scope/impact.
To further enhance the manuscript's comprehensiveness, I suggest incorporating additional examples of amyloid-like toxic assemblies relevant to both PD and cancer, addressing the current limitation of relying on a phenylketonuria reference. Additionally, providing a more thorough explanation of the relatively new and emerging concept of the α-synuclein-independent pathway of β-synuclein would align with its growing importance in the field.
In conclusion, this manuscript offers a valuable contribution to the understanding of the intriguing PD-cancer connection. By addressing the aforementioned suggestions, the authors can elevate the manuscript's clarity and potential impact within the field, paving the way for future advancements in therapeutic strategies for both diseases.
Minor point:
- Consider incorporating a diagram or infographic illustrating the key similarities and differences between PD and cancer for the main and interacted pathogenic factors, such as TFs, inflammation and a-synuclein. This visual representation could effectively highlight the shared signaling pathways and potential therapeutic targets.
- correct the typo at line 249 (5.1a-.Synuclein).
Author Response
I encourage the authors to address concerns regarding the flow and coherence of section 4.1. Ensuring smooth transitions between sentences and ideas will greatly facilitate comprehension. Furthermore, section 4.2 would benefit from a more detailed exploration of the potential role of apoptosis-promoting mutations, accompanied by comprehensive and relevant references to support the statements.
Thank you. We made smooth transitions between sections and added the following text and references after Section 4.1.
“MAMs participate in many basic biological processes, including calcium and lipid homeostasis, autophagy, inflammation, and apoptosis. MAMs play an essential rolein in maintaining the normal cellular functions, and the disturbance of MAMs by mutations or chemical agents may cause neurodegeneration and cancer development [33].”
We also added the text and reference after Section 4.2 (lines 148-156):
Gong et al. (2017) showed that mutations of the PARK2 gene can promote both PD and cancer...
Similarly, comprehensive references are essential to substantiate claims in section 4.4, unless solely based on Schernhammer and coauthors' work.
We added to Section 4.4. the following text (lines 200-205):
Hardeland (2012) noted that a strong reduction of circulating melatonin occurred in neurological disorders and cancer. An important role is played by the association between the mutual relationship between circadian oscillators and melatonin secretion. Importantly, the readjustment of rhythms controlled by melatonin and its synthetic analogs may be used to improve the course of the circadian rhythm-dependent disorders.
I commend the authors for recognizing the intriguing potential of shared pathogenetic factors and mediators. Two areas, in particular, present fertile ground for further exploration: transcription factors and chronic inflammation.
Section 4.5 highlights the importance of transcription factors, particularly TFEB. As the authors acknowledge, TFEB activation has been shown to stimulate cellular regeneration, offering a glimmer of hope for neurodegenerative diseases like PD. However, its potential relevance to cancer therapy required a more detail explanation. I recommend delving deeper into its potential relevance for cancer therapy by exploring its interplay with cancer-suppressing pathways, such as autophagy, providing specific examples and pertinent references.
We added the following text in response to TFEB comments and suggestion:(lines 219-230) beginning with: Nabar et al. (2021) [57] noted that LRRK2 is an upstream activator of TFEB which is a host defense transcription factor and the master transcriptional regulator of the autophagy/lysosome machinery....
In Moor et al. study (2017) [58] the authors pointed to an abnormal regulation of autophagy associated with the aggregation of α-synuclein….
Chronic inflammation, while mentioned in section 4.8, warrants a more comprehensive examination. Expanding on this theme by providing a more comprehensive characterization can offer valuable insights into their shared pathogenic mechanisms and pave the way for novel therapeutic strategies.
Thank you. We added data on chronic inflammation to Sections 4.1; 4.2; 4.7 and 4.8 (Fung et al.) Shared pathogenic mechanisms are now also discussed in Sections 3; 4.5; 5.3; 9 and Table1
To further enhance the manuscript's comprehensiveness, I suggest incorporating additional examples of amyloid-like toxic assemblies relevant to both PD and cancer.
Thank you, we added Figure 1 and Figure legend to better illustrate this issue (“amyloid-like toxic assemblies relevant to human diseases”).
Additionally, providing a more thorough explanation of the relatively new and emerging concept of the α-synuclein-independent pathway of β-synuclein would align with its growing importance in the field.
Thank you for this comment. In response we added the following text and references in section 7.1 (lines 473-475):
β-Synuclein is implicated in age-related as well as pathophysiological conditions and regulates p53-mediated and Akt-independent apoptosis [109]. Accumulating new data points to an emerging concept of the α-synuclein-independent pathway of β-synuclein [110,111]
Minor point:
- Consider incorporating a diagram or infographic
We added Figure 1 to illustrate α-synuclein and other amyloids gain-of-function which may accelerate neurodegeneration and modulate diseases progression.
2.correct the typo at line 249 (5.1a-.Synuclein).
Thank you, it is corrected.